# Effect of Dietary Patterns on Vascular Aging Using the Brachial–Ankle Index

**DOI:** 10.3390/nu16234229

**Published:** 2024-12-06

**Authors:** Inés Llamas-Ramos, Rocío Llamas-Ramos, María Cortés-Rodríguez, Emiliano Rodríguez-Sánchez, Luis García-Ortiz, Manuel A. Gómez-Marcos, Marta Gómez-Sánchez, Leticia Gómez-Sánchez

**Affiliations:** 1Faculty of Nursing and Physiotherapy, Universidad de Salamanca, 37007 Salamanca, Spain; inesllamas@usal.es (I.L.-R.); rociollamas@usal.es (R.L.-R.); 2Instituto de Investigación Biomédica de Salamanca (IBSAL), 37007 Salamanca, Spain; emiliano@usal.es; 3Primary Care Research Unit of Salamanca (APISAL), Health Centre of San Juan. Av. Portugal 83, 2° P, 37005 Salamanca, Spain; lgarciao@usal.es (L.G.-O.); martagmzsnchz@gmail.com (M.G.-S.); leticiagmzsnchz@gmail.com (L.G.-S.); 4University Hospital of Salamanca, 37007 Salamanca, Spain; 5Department of Statistics, Universidad de Salamanca, 37008 Salamanca, Spain; 6Department of Hematology, University Hospital of Salamanca, 37008 Salamanca, Spain; 7Primary Healthcare Management, Castilla y León Regional Health Authority (SACyL), 37007 Salamanca, Spain; 8Department of Medicine, Universidad de Salamanca, 37007 Salamanca, Spain; 9Red de Investigación en Cronicidad, Atención Primaria y Promoción de la Salud (RICAPPS), 37005 Salamanca, Spain; 10Department of Biomedical and Diagnostic Sciences, Universidad de Salamanca, 37007 Salamanca, Spain; 11Home Hospitalization Service, Marqués of Valdecilla University Hospital, s/n, 39008 Santander, Spain; 12Emergency Service, University Hospital of La Paz P. of Castellana, 261, 28046 Madrid, Spain

**Keywords:** vascular function, Mediterranean diet, healthy aging, brachial–ankle index

## Abstract

The Mediterranean diet (MD) plays an important role in delaying vascular aging. The main objective of this study was to analyze the association between adherence to the MD and vascular aging estimated with brachial–ankle pulse wave velocity (ba-PWV) in a Spanish population sample and the differences by sex. Methods: Cross-sectional descriptive study. A total of 3437 subjects from the EVA, MARK and EVIDENT studies participated. The ba-PWV was assessed with the Vasera VS-1500^®^ device. Vascular aging was classified as healthy vascular aging (HVA), normal vascular aging (NVA) and early vascular aging (EVA) and adherence to the MD was assessed with the Mediterranean Diet Adherence Screener questionnaire. Results: The mean age was 60.15 ± 9.55 (60.09 ± 9.71 in women; 60.20 ± 9.43 in men). Overall, MD adherence was observed in 48.0% of subjects (40% in women; 54% in men). The mean value of ba-PWV was 14.38 ± 2.71 (women 14.24 ± 2.89; men 14.49 ± 2.56). In multinomial logistic regression a positive association was found overall between HVA and NVA (OR = 1.751, 95% CI: 1.411–2.174, *p* < 0.001) and between HVA and EVA (OR = 1.501, 95% CI:1.295–1.740, *p* < 0.001); in women between HVA and NVA (OR = 2. 055, 95% CI:1.456–2.901, *p* < 0.001) and between HVA and EVA (OR = 1.413, 95% CI:1.124–1.776; *p* = 0.003); and in men between HVA and NVA (OR = 1.551, 95% CI: 1.175–2.047, *p* = 0.002) and between HVA and EVA (OR = 1.549, 95% CI: 1.275–1.882; *p* < 0.001). Conclusions: The results of this work indicate that greater adherence to the MD is associated with healthier vascular aging.

## 1. Introduction

In modern societies, life expectancy has increased significantly, leading to a higher prevalence of diseases associated with aging, such as cardiovascular diseases [1,2,3]. Cardiovascular diseases continue to be the leading cause of death worldwide, with an estimated 24 million deaths per year by 2030 [4]. For that reason, the study of vascular aging is essential to find effective preventive measures [5]. One of the main indicators of vascular aging is arterial stiffness [6]. Arterial stiffness is the resistance of arteries to deformation in response to pressure/flow changes with each heartbeat. Arterial stiffness is a noninvasive marker of early-to-late stages of atherosclerosis and is considered a surrogate for cardiovascular disease [7]. Among the most common ways to measure arterial stiffness is pulse wave velocity (PWV) [3], which assesses the speed with which a pulse wave travels through the arteries; this velocity increases as blood vessels become stiffer so that a higher PWV is associated with a higher cardiovascular risk [2]. Among the indices that evaluate PWV is the brachial–ankle index (ba-PWV), which measures the velocity of blood displacement between the brachial artery and the tibial artery using an oscillometric method; it is a simple, noninvasive measurement of peripheral arterial stiffness, is considered a good indicator of vascular health [3,8] and shows a strong correlation with age [3]. Knowing the state of vascular health in the adult population is of vital importance since any dysfunction can have harmful effects on health [8].

Vascular aging is a constantly evolving area of study, since cardiovascular diseases are closely related to the aging of blood vessels. In recent years, progress has been made in understanding the molecular mechanisms, cellular signals and genetic factors involved in vascular aging, which is a complex process in which genetic, environmental and lifestyle factors interact [9]. The study of vascular aging also focuses on preventing or slowing down the negative effects, the objective being to preserve health and vitality, these concepts giving rise to the term “healthy aging” [5]. Early diagnosis and timely therapeutic interventions are the key to success in delaying or preventing aging [3,10]. To promote healthy aging, there are different strategies proposed such as implementing regular physical exercise [11,12], controlling risk factors [13] or maintaining healthy eating habits [14,15,16], especially with suboptimal diets and lifestyles [12,17,18]. This has been reflected in the literature, where a comprehensive lifestyle intervention over 30 years was associated with a reduction in risk factors by 80% [13]. It is estimated that about 11 million premature deaths can be attributed to unhealthy dietary factors [19]. Combining diet with other healthy lifestyles could increase life expectancy by 8-to-10 years [17].

Diet seems to have a close relationship with healthy aging; in particular, adherence to the Mediterranean diet (MD) [20] may be associated with a lower cardiovascular risk [21]. In addition to the decrease in the incidence of cardiovascular disease, there are risk factors such as obesity, hypertension, metabolic syndrome and dyslipidemia for which the MD has shown numerous benefits. Adherence to the MD has reduced mortality, especially that caused by cardiovascular events. Of note is the lower incidence of diabetes and neurodegenerative disorders, particularly Alzheimer’s disease [22]. The MD [20] exhibits dietary patterns typical of populations located near the Mediterranean Sea and is particularly associated with olive-growing areas. In addition, it is associated with decreased rates of chronic diseases, especially cardiovascular and metabolic diseases associated to a greater extent with obesity, which influences a longer life expectancy [16,23]. This diet is characterized by a high consumption of olive oil and plant foods (vegetables, legumes, potatoes, fruits, bread and other cereals, seeds and nuts) and consumption of seasonal, fresh and minimally processed products. It also recommends a moderate consumption of fish and dairy products such as yogurts and cheeses and a moderate–low consumption of poultry. Regarding eggs, no more than four per week are recommended [20]. Regarding caloric intake of fats, the MD should not exceed 30% of intake, with saturated fats accounting for 8–10% [24,25]. Therefore, lifestyle factors such as diet can modify the aging process [23,26]. The MD is considered to be one of the healthiest diets, highlighting its influence in improving vascular aging [1,16]. In addition to the benefits noted, the MD has been effective in increasing HDL cholesterol and reducing weight, body mass index, waist circumference, total cholesterol and inflammatory parameters versus other diets [27]. Notably, adoption of this type of diet for at least one year can reduce systolic and diastolic pressures [28,29]. In the review study carried out by Finicelli et al. in 2022 they observed the benefits of the MD in the prevention of metabolic problems such as obesity, type 2 diabetes mellitus or cardiovascular complications [30]. However, they highlight that adherence to this type of diet decreases during the period of self-monitoring, so that following the recommendations and complying with the dietary pattern is a fundamental aspect [30]. Good nutrition [1], lifestyles [31], physical activity [32,33] and psychological balance [34] are the pillars that help to have a healthy aging [1]. Therefore, the aim of this study is to analyze the association between adherence to the MD with vascular aging estimated with ba-PWV and differences by sex in a sample of a Caucasian population without a history of previous cardiovascular disease.

## 2. Materials and Methods

### 2.1. Study Design

Descriptive and observational cross-sectional study. The data analyzed correspond to the data recruited in the MARK [35] (NCT01428934), EVIDENT [36] (NCT0108308) and EVA [37] (NCT02623894) studies.

### 2.2. Ethical Considerations

The three studies selected have been approved by the ethics committee of the Salamanca area. The MARK study [35] has the code PI10/02043, the EVIDENT study [36] has the code PI83/06/2018 and the EVA study [37] has the codes PI15/01039 and PI20/10569. The objectives of this study were explained and participants signed consent before being included in the studies. The norms established in the Declaration of Helsinki and the WHO standards for observational studies were always followed. Data confidentiality was guaranteed throughout the process.

### 2.3. Study Population

The three studies were conducted in primary care centers. The three studies included 4103 subjects between 35 and 75 years of age. In this work 669 subjects have been excluded because not all the variables necessary to carry out this work have been recorded. Data from 3437 subjects were analyzed. In the MARK study [35], by random sampling of subjects consulting in six primary care centers in three autonomous communities, a total of 2511 individuals were recruited. They were between 35 and 75 years of age and had an intermediate cardiovascular risk as a 10-year coronary risk between 5% and 15% estimated with the adapted Framingham risk [38], a 10-year vascular mortality risk between 1% and 5% estimated with the equation for the European [39] or a moderate risk according to the European Society of Hypertension criteria for the treatment of arterial diseases [40]. Exclusion criteria included the following: having end-stage disease, being institutionalized at the time of the visit or having a history of atherosclerosis. Recruitment was performed from July 2011 to June 2013. A total of 2505 subjects were included in the analysis of this work.

The EVIDENT study [36] had 1104 subjects selected by random sampling from a primary care center. Exclusion criteria were known coronary or cerebrovascular atherosclerotic disease; heart failure; moderate or severe chronic obstructive pulmonary disease; musculoskeletal disease that limited walking; advanced respiratory, renal or hepatic disease; severe mental disease; and treated oncological disease diagnosed in the last 5 years. Recruitment was performed from January 2014 to September 2016. A total of 432 subjects were included in the analysis of this work.

In the EVA study [37], using random sampling with replacement stratified by age groups (35, 45, 55, 65 and 75 years) and sex, we selected 501 subjects without previous cardiovascular disease receiving care in five urban health centers. We selected 100 subjects for each group—with the subjects being between 35 and 75 years of age and the groups comprising an equal amount of each sex—from a reference population of 43946 included in the individual health card database. Inclusion criteria were an age between 35 and 75 and an absence of previous cardiovascular disease. Exclusion criteria were terminally ill subjects, an inability to travel to health centers, a history of cardiovascular disease, a glomerular filtration rate < 30 mL/min/1.73 m^2^, chronic inflammatory disease or an acute inflammatory process in the last 3 months or being on estrogen, testosterone or growth hormone treatment. Recruitment was performed from June 2016 to November 2017. A total of 500 subjects were included in the analysis of this work.

Figure 1 shows the subjects included and excluded from each of the three studies.

### 2.4. Variables and Measuring Instruments

#### 2.4.1. Vascular Aging

The following steps were followed to define vascular aging. First, the values established by the EVA group as cut-off points estimated with the ba-PWV for both men and women were taken into account [41]. Subjects with values ≤ 25th percentile were classified as HVA, subjects with values between the 25th percentile and 75th percentile were classified as NVA and subjects with values ≥ 75th percentile were classified as EVA. Secondly, subjects diagnosed with diabetes who belonged to the HVA group were reclassified as NVA [31].

#### 2.4.2. Mediterranean Diet

To determine adherence to the MD, we used the questionnaire validated in the Spanish population, which was used in the PREDIMED study [42]. The Mediterranean Diet Adherence Screener (MEDAS) questionnaire consists of 14 items, 12 of which ask about the frequency of food consumption and 2 of which ask about the eating habits of the Spanish population. Each of the questions can be scored with 0 or 1 point, the final score ranges from 0 to 14 points and adherence to the MD is considered if the score is higher than the median (7 or more points).

To obtain the score: (a) consume olive oil as the main cooking fat; (b) consume four or more tablespoons of olive oil daily (one tablespoon = 13. 5 g); (c) consume two or more servings of vegetables; (d) consume three or more pieces of fruit; (e) consume less than one serving of red or processed meat; (f) consume less than one serving of animal fat; (g) consume less than one 100 mL cup of sweetened beverages; (h) eat white meat in a greater proportion than red meat; (i) consume seven or more glasses of wine per week; (j) consume three or more servings of legumes; (k) consume three or more servings of fish; (l) consume three or more servings of nuts or dried fruits; (m) consume two or more servings of fried foods; and (n) consume less than two baked goods. These were evaluated with 1 point [42].

#### 2.4.3. Vascular Function

The function was estimated using the brachial–ankle index (ba-PWV) with the VaSera VS-1500^®^ device (Fukuda Denshi, Tokyo, Japan).

The following equation was used to estimate ba-PWV: ba-PWV = ((0.5934 × height (cm) + 14.4724))/tba, where tba is the transmission time of the waves to the ankle [43,44]. For this study, the mean value of the two extremities of ba-PWV was taken into account.

#### 2.4.4. Anthropometric Measurements

Weight, height and blood pressure were taken into account. Hypertension was considered present if the subjects were taking antihypertensive drugs or if their blood pressure values were higher than 140/90 mmHg. They were considered to have type 2 diabetes mellitus if they were taking hypoglycemic agents or if their fasting plasma glucose values were higher than 126 mg/dL or higher than 6.5% in the case of HbA1c. Dyslipidemia was considered present if they were taking lipid-lowering drugs or had fasting total cholesterol values higher than 240 mg/dL, low-density lipoprotein cholesterol (LDLc) higher than 160 mg/dL, high-density lipoprotein cholesterol (HDLc) lower than 40 mg/dL in men and lower than 50 mg/dL in women or triglyceride values higher than 150 mg/dL. Obesity was defined using a BMI value above 30 kg/m^2^ [40].

In addition, venous blood samples were taken from all participants between 8:00 a.m. and 9:00 a.m.

All measurements were performed by primary care professionals.

### 2.5. Statistical Analysis

Continuous variables are presented as means ± standard deviations and categorical variables as numbers or percentages. Chi-square tests for percentages and Student’s t-tests for continuous variables were used to compare men and women. The ANOVA test was used to compare means between the three aging groups (EVA, HVA and NVA). Multinomial logistic regression was performed to analyze the association between adherence to the MD and arterial stiffness, measured through ba-PWV and categorized into three levels according to the 25th and 75th percentile cut-off points. The MD was assessed using a numerical scale and was included as a main predictor variable. Regression was adjusted for age, sex and consumption of antidiabetic, lipid-lowering and hypertensive drugs. Age, sex and consumption of lipid-lowering, hypolipidemic, antihypertensive and lipid-lowering drugs were used as adjustment variables. All analyses were performed by gender and globally. SPSS Statistics for Windows, version 28.0 (IBM Corp, Armonk, NY, USA) was used. A value of *p* < 0.05 was considered as the limit of statistical significance.

## 3. Results

### 3.1. Baseline Characteristics

The total sample consisted of 3437 subjects: 500 subjects belonged to the EVA study, 2505 subjects participated in the MARK study and 432 belonged to the EVIDENT study (Figure 1). An amount of 43% of the sample consisted of women (n = 1468) and the mean age was 60.15 ± 9.55 (60.09 ± 9.71 for women and 60.20 ± 9.43 for men), *p* = 0.728. The score obtained in the MD was 5.82 ± 2.03 overall (6.03 ± 1.98 for women and 5.66 ± 2.06 for men), *p* < 0.001. The mean ba-PWV value was 14.38 ± 2.71 (14.24 ± 2.89 for women and 14.49 ± 2.56 for men), *p* = 0.007 (Table 1). This table also shows the values of the drugs used in the treatment of dyslipidemia, diabetes mellitus and hypertension, in addition to the values obtained for cardiovascular risk factors in the overall population and by sex.

The graphical representation of the correlations is shown in the heat map in Figure 2.

### 3.2. Characteristics of the Subjects in Relation to Vascular Aging and the Mediterranean Diet

Table 2 shows the differences in the total sample in the MD, cardiovascular risk factors and drugs used in the subjects according to their degree of aging. An amount of 14% of the sample (n = 475) was classified as HVA, 45% (n = 1559) was classified as NVA and the remaining 41% (n = 1403) was classified as EVA (Figure 3). The mean age for subjects in the HVA group was 59.39 ± 10.34 (60.28 ± 9.30 for men and 57.93 ± 11.73 for women), for the NVA group it was 60.33 ± 9.78 (60.34 ± 9.60 for men and 60.31 ± 10.04 for women) and for the EVA group it was 60.22 ± 8.98 (60.02 ± 9.29 for men and 60.47 ± 8.59 for women). For the ba-PWV variable, a lower velocity was observed for the HVA group with values of 11.60 ± 1.34 (11.91 ± 1.17 for men and 11.08 ± 1.45 for women) than in the NVA group with values of 13.45 ± 1. 69 (13.59 ± 1.43 for men and 13.25 ± 1.99 for women) and the EVA group with values of 16.36 ± 2.57 (16.52 ± 2.49 for men and 16.18 ± 2.67 for women) being significant in all cases (*p* < 0.001).

Table 3 shows the differences by sex of the sample in the MD, cardiovascular risk factors and drugs used in the subjects according to their degree of aging.

Figure 3 shows the classification according to the degree of aging and origin of the subjects in each of the three studies.

### 3.3. Association Between ba-PWV and the MD Overall and by Sex Determined by Multinomial Logistic Regression

A multinomial logistic regression was performed to determine the association between ba-PWV and the MD, finding a significant relationship both globally and by sex. Globally, a strong association was observed between HVA and NVA (OR = 1.751, 95% CI:1.411–2.174, *p* < 0.001), as was the association between HVA and EVA (OR = 1.501, 95% CI:1.295–1.740, *p* < 0.001). Women showed an even stronger association between HVA and NVA (OR = 2.055, 95% CI:1.456–2.901, *p* < 0.001), and also with EVA (OR = 1.413, 95% CI:1.124–1.776, *p* = 0.003). The association was also significant for men, although slightly more moderate: HVA vs. NVA (OR = 1.551, 95% CI:1.175–2.047, *p* = 0.002), and HVA vs. EVA (OR = 1.549, 95% CI:1.275–1.882, *p* < 0.001) (Table 4).

## 4. Discussion

This study shows the association between adherence to the MD and vascular aging analyzed with the brachial–ankle index (ba-PWV) in a sample of 3437 Caucasian subjects. The most important results found in this work are a strong positive association between HVA and NVA and between HVA and EVA both overall and by sex.

It is not defined how adherence to the MD is affected by sex. Bédard et al. wanted to test for sex-related differences in the cardiovascular response associated with the diet and did so by comparing 38 men and 32 premenopausal women. No diet-induced LDL cholesterol reduction was found so they concluded that adherence to the MD reduces cardiovascular risks independently of sex-related factors [45]. Our study is in agreement with a recent review and meta-analysis: although the results were not significant, it seems that the MD is more favorable for women as they adhere better and drop out less than men [46]. Susanto et al. argued that sex-based factors affecting outcomes such as weight loss should be examined separately for men and women in order to avoid overlooking differences that could contribute to improved outcomes [47].

On the other hand, there are several studies showing the benefits of the MD. Among them we can mention the NU-AGE study, with 1296 participants, which evaluated whether the MD specifically targeting people over 65 years of age could be effective in shifting the intake of older adults towards a healthy diet [48]. Similarly, Jennings et al. in 2019 found that the MD was effective in improving cardiovascular health, with significant reductions in blood pressure and arterial stiffness [49]. In recent years, followers of other diets are increasing. The combination of the Atlantic diet and physical activity were associated with a lower cardiovascular risk and lower pulse wave velocity values [50]. On the other hand, Sandri et al. evaluated the diet of 19211 citizens in Spain, finding that n = 1638 followed a plant-based diet. They tend to follow healthier life patterns and consume less unhealthy foods; however, these types of diets should be supervised by health professionals [51].

The study by Assmann et al. with a sample of 3012 participants aged 45 to 65 years investigated the association between adherence to the MD in midlife and healthy aging, finding that a high adherence to the MD is favorable in midlife for maintaining good general health during aging [52]. These works are in line with the results shown in this work: the multinomial logistic regression results of the present study show that adherence to the MD is associated with lower arterial stiffness, evidenced by the values obtained in all comparisons. This association was observed to a greater extent among individuals belonging to the HVA and NVA groups and between HVA and EVA. Moreover, the association was greater in women, especially between HVA and NVA (OR = 2.055 vs. OR = 1.551 in men). It can be deduced that the MD is not only associated with a more favorable risk profile but may also be a crucial factor in improving vascular health in the Spanish adult population. Although the values were slightly better in women, significant associations were observed in both sexes, which suggests that better adherence to the MD may act as a protector against vascular aging. Similarly, in the EPIC study developed in Spain by Guallar-Castillón et al. it was found that people with higher MD scores had a lower long-term risk of coronary heart disease [53].

It should be noted that studies have also been found in the literature that implement different strategies with the aim of improving health by modifying lifestyles, including adherence to the MD, such as combining a yoga program with the MD, which obtained good results in the intervention group [54], or including new technologies such as a smartphone and smartband, although in the latter case there were no changes in lifestyle related to physical activity or eating habits [55]. There are several reviews that try to compile the evidence on MD adherence in different diseases such as cardiovascular, metabolic or cancer [22,30], or even differentiating the results by sex [47,48]. In all of them the benefits of this type of diet are demonstrated; however, more studies are needed to specify the associations between the different strategies that are carried out, as well as to determine to what extent they are effective or to establish the superiority of some over others in improving vascular health in this population.

The present study has a number of limitations. Firstly, the sample is made up of patients without previous cardiovascular disease, so the results cannot be extrapolated to other parts of the population. On the other hand, the sample analyzed is part of three studies in which the percentage of subjects included is not the same, since the MARK study contributed the greatest number of subjects (2505) compared to EVA (501) or EVIDENT (432). On the other hand, the data were extracted at a single point in time, so it is a cross-sectional study that does not allow us to know cause and effect. However, it is worth highlighting the strengths of this study: it is a study with a large sample recruited from primary care clinics in Spain that investigated adherence to the MD in a population of subjects aged 35 to 75 years to determine its association with healthy aging. This study has had a protocolized design that ensures the correct measurement and evaluation of the subjects. The findings of this study underscore the relevance of healthy vascular aging in the prevention of vascular dysfunction, with implications for both public health and gender-differentiated intervention strategies.

## 5. Conclusions

The results of this study indicate that greater adherence to the MD is associated with healthier vascular aging. This suggests the importance of promoting the MD as a public health intervention strategy to prevent vascular aging and its consequences. Therefore, health policy strategies are needed to promote the Mediterranean-style diet to the entire population. All this will help to reduce the burden on health care systems, preventing the development of chronic diseases and improving vascular aging.

## Figures and Tables

**Figure 1 nutrients-16-04229-f001:**
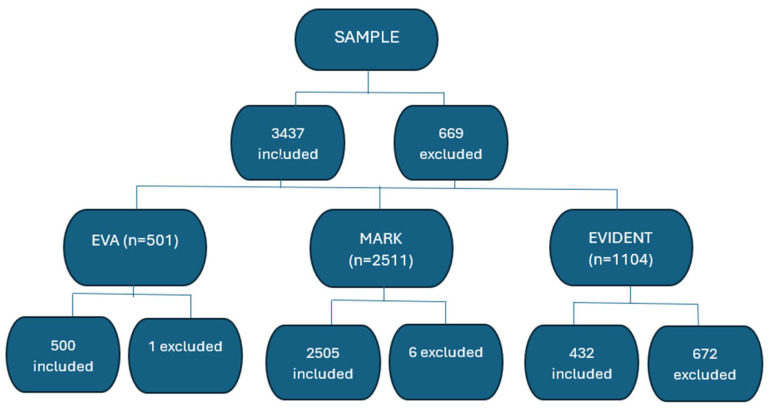
This flow chart describes to which study each of the 3437 subjects included in the random sampling selection belonged.

**Figure 2 nutrients-16-04229-f002:**
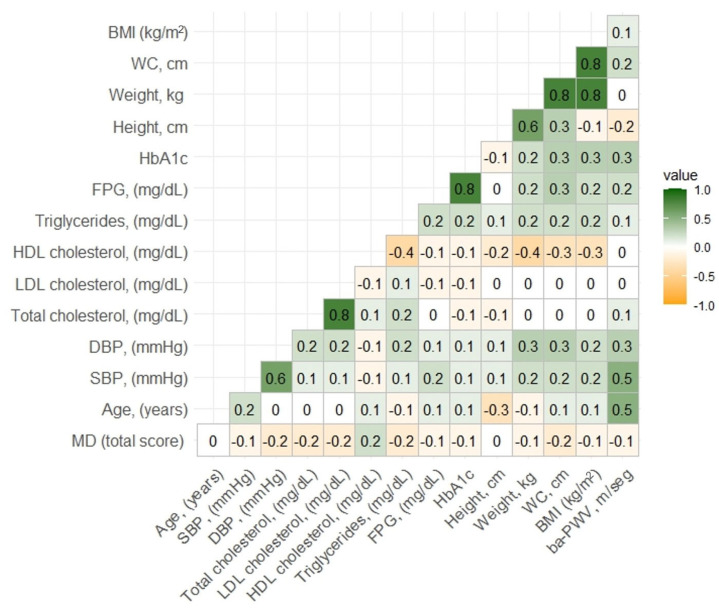
Correlation matrix between continuous variables.

**Figure 3 nutrients-16-04229-f003:**
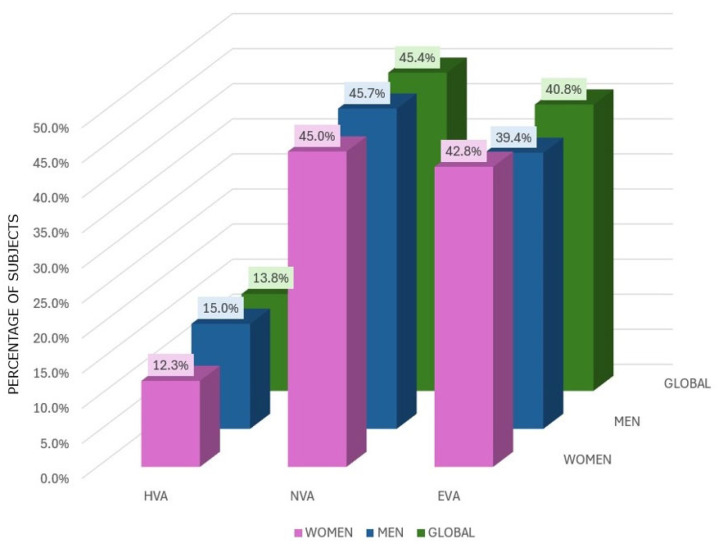
Global proportion of subjects with HVA, NVA and EVA.

**Table 1 nutrients-16-04229-t001:** General characteristics of the subjects included overall and by sex.

	Global (n = 3437)	Women (n = 1468)	Men (n = 1969)	*p*-Value
Mediterranean Diet				
MD (total score)	5.82 ± 2.03	6.03 ± 1.98	5.66 ± 2.06	<0.001
Adherence to MD, n (%)	1650(48.0)	588(40.1)	1062(53.9)	<0.001
Conventional risk factors				
Age, (years)	60.15 ± 9.55	60.09 ± 9.71	60.20 ± 9.43	0.728
SBP, (mmHg)	133.45 ± 19.53	128.72 ± 20.55	136.99 ± 17.93	<0.001
DBP, (mmHg)	82.07 ± 10.97	79.64 ± 10.84	83.89 ± 10.71	<0.001
Hypertension, n (%)	2943(85.6)	1309(89.2)	1634(83)	<0.001
Antihypertensive drugs, n (%)	1573(45.8)	654(44.6)	919(46.7)	0.217
Total cholesterol, (mg/dL)	216.42 ± 41.11	220.54 ± 42.47	213.34 ± 39.80	<0.001
LDL cholesterol, (mg/dL)	133.09 ± 35.18	132.53 ± 35.94	133.50 ± 34.60	0.426
HDL cholesterol, (mg/dL)	52.54 ± 14.50	57.27 ± 15.77	49.01 ± 12.34	<0.001
Triglycerides, (mg/dL)	133.38 ± 83.74	122.63 ± 68.62	141.42 ± 92.67	<0.001
Dyslipidemia, n (%)	2487(72.4)	1076(73.3)	1411(71.7)	0.289
Lipid-lowering drugs. n (%)	974(28.3)	429(29.2)	545(27.7)	0.320
FPG, (mg/dL)	101.75 ± 31.82	100.24 ± 32.30	102.88 ± 31.42	0.016
HbA1c	5.94 ± 1.05	5.93 ± 1.07	5.94 ± 1.04	0.680
Diabetes mellitus, n (%)	759(22.1)	306(20.8)	453(23.0)	0.131
Hypoglycaemic drugs, n (%)	576(16.8)	234(15.9)	342(17.4)	0.267
Weight, kg	77.52 ± 14.65	70.20 ± 13.31	82.98 ± 13.15	<0.001
Height, cm	164.59 ± 9.46	157.11 ± 6.67	170.17 ± 7.08	<0.001
BMI, (kg/m^2^)	28.56 ± 4.53	28.47 ± 5.30	28.62 ± 3.86	0.367
WC, cm	98.65 ± 12.07	94.33 ± 12.91	101.87 ± 10.30	<0.001
Obesity, n (%)	1086(31.6)	495(33.7)	591(30.0)	0.021
Arterial stiffness				
ba-PWV, m/s	14.38 ± 2.71	14.24 ± 2.89	14.49 ± 2.56	0.007

Values are means ± standard deviations for continuous data and number and proportions for categorical data. MD: Mediterranean diet; SBP: systolic blood pressure; DBP: diastolic blood pressure; LDL: low-density lipoprotein; HDL: high-density lipoprotein; FPG: fasting plasma glucose; HbA1c: glycosylated hemoglobin; BMI: body mass index; WC: waist circumference; *p*-value: differences between men and women.

**Table 2 nutrients-16-04229-t002:** Characteristics of the subjects included with and without healthy vascular aging.

	HVA(n = 475)	NVA(n = 1559)	EVA(n = 1403)	*p*-Value	
Mediterranean Diet					
MD (total score)	6.20 ± 2.10	5.98 ± 2.02	5.51 ± 1.99	<0.001	
Adherence to MD, n (%)	267(56.2)	805(51.6)	578(41.2)	<0.001	
Conventional risk factors					
Age, (years)	59.39 ± 10.34	60.33 ± 9.78	60.22 ± 8.98	0.207	
SBP, (mmHg)	121.03 ± 15.38	129.98 ± 18.57	141.52 ± 18.45	<0.001	
DBP, (mmHg)	76.27 ± 10.19	80.23 ± 10.03	86.08 ± 10.80	<0.001	
Hypertension, n (%)	455(95.8)	1382(88.6)	1106(78.8)	<0.001	
Antihypertensive drugs, n (%)	161(33.9)	708(45.4)	704(50.2)	<0.001	
Total cholesterol, (mg/dL)	211.92 ± 36.34	213.49 ± 41.34	221.18 ± 41.91	<0.001	
LDL cholesterol, (mg/dL)	131.25 ± 31.60	131.13 ± 35.51	135.90 ± 35.80	<0.001	
HDL cholesterol, (mg/dL)	54.16 ± 13.52	52.99 ± 15.00	51.49 ± 14.17	<0.001	
Triglycerides, (mg/dL)	112.48 ± 55.29	128.77 ± 77.38	145.61 ± 95.84	<0.001	
Dyslipidemia, n (%)	286(60.2)	1098(70.4)	1103(78.6)	<0.001	
Lipid-lowering drugs. n (%)	98(20.6)	458(29.4)	418(28.8)	<0.001	
FPG, (mg/dL)	88.23 ± 11.41	99.13 ± 26.61	109.06 ± 39.17	<0.001	
HbA1c	5.46 ± 0.36	5.86 ± 0.90	6.18 ± 1.27	<0.001	
Diabetes mellitus, n (%)	0(0.0)	340(21.8)	419(29.9)	<0.001	
Hypoglycaemic drugs, n (%)	0(0.0)	256(16.4)	320(22.8)	<0.001	
Weight, kg	77.99 ± 14.49	77.13 ± 14.86	77.93 ± 14.47	0.356	
Height, cm	166.71 ± 9.44	164.64 ± 9.38	163.82 ± 9.45	<0.001	
BMI, (kg/m^2^)	28.02 ± 4.40	28.40 ± 4.74	28.90 ± 4.31	<0.001	
WC, cm	96.86 ± 11.93	98.14 ± 12.56	99.83 ± 11.44	<0.001	
Obesity, n (%)	123(25.9)	484(31.0)	479(34.1)	<0.001	
Arterial stiffness					
ba-PWV, m/s	11.60 ± 1.34	13.45 ± 1.69	16.36 ± 2.57	<0.001	

Values are means ± standard deviations for continuous data and number and proportions for categorical data. HVA: healthy vascular aging; NVA: normal vascular aging; EVA: early vascular aging; MD: Mediterranean diet; SBP: systolic blood pressure; DBP: diastolic blood pressure; LDL: low-density lipoprotein; HDL: high-density lipoprotein; FPG: fasting plasma glucose; HbA1c: glycosylated hemoglobin; BMI: body mass index; WC: waist circumference; *p*-value: differences between groups.

**Table 3 nutrients-16-04229-t003:** Characteristics of the subjects included with and without healthy vascular aging by sex.

	HVA(n = 475)		NVA(n = 1559)		EVA(n = 1403)		*p*-Value	
	Men (n = 295)	Women (n = 180)	Men (n = 899)	Women (n = 660)	Men (n = 775)	Women (n = 628)	Men	Women
Mediterranean Diet								
MD (total score)	5.92 ± 2.11	6.68 ± 2.01	5.85 ± 2.07	6.16 ± 1.93	5.33 ± 1.99	5.72 ± 1.97	<0.001	<0.001
Adherence to MD, n (%)	170(57.6)	97(53.9)	524(58.3)	281(42.6)	368(47.5)	210(43.4)	<0.001	<0.001
Conventional risk factors								
Age, (years)	60.28 ± 9.30	57.93 ± 11.73	60.34 ± 9.60	60.31 ± 10.04	60.02 ± 9.29	60.47 ± 8.59	0.771	0.025
SBP, (mmHg)	125.70 ± 14.16	113.38 ± 14.23	133.37 ± 15.63	125.37 ± 21.11	145.48 ± 17.95	136.63 ± 17.90	<0.001	<0.001
DBP, (mmHg)	78.46 ± 9.14	72.68 ± 10.80	82.06 ± 9.45	77.75 ± 10.27	88.06 ± 11.13	83.63 ± 9.85	<0.001	<0.001
Hypertension, n (%)	276(93.6)	179(99.4)	781(86.9)	601(91.1)	577(74.5)	529(84.2)	<0.001	<0.001
Antihypertensive drugs, n (%)	107(36.3)	54(30.0)	427(47.5)	281(42.6)	385(49.7)	319(50.8)	<0.001	<0.001
Total cholesterol, (mg/dL)	214.11 ± 35.88	208.32 ± 36.90	209.48 ± 40.55	218.96 ± 41.82	217.52 ± 39.95	225.71 ± 43.83	<0.001	<0.001
LDL cholesterol, (mg/dL)	135.43 ± 31.75	124.35 ± 30.19	131.07 ± 34.89	131.20 ± 36.36	135.60 ± 35.19	136.26 ± 36.56	0.018	<0.001
HDL cholesterol, (mg/dL)	50.66 ± 11.29	59.95 ± 14.87	48.69 ± 12.01	58.85 ± 16.61	48.75 ± 13.05	54.85 ± 14.78	0.028	<0.001
Triglycerides, (mg/dL)	122.33 ± 60.43	96.39 ± 40.98	135.16 ± 78.53	120.09 ± 79.97	156.03 ± 113.67	132.84 ± 65.76	<0.001	<0.001
Dyslipidemia, n (%)	187(63.4)	99(55.0)	639(71.1)	459(69.5)	585(75.5)	518(82.5)	<0.001	<0.001
Lipid-lowering drugs. n (%)	64(21.7)	34(18.9)	266(29.6)	192(29.1)	215(27.7)	203(32.3)	0.031	0.002
FPG, (mg/dL)	90.99 ± 11.13	85.29 ± 10.99	100.86 ± 26.26	96.78 ± 26.94	109.78 ± 39.49	108.17 ± 38.79	<0.001	<0.001
HbA1c	5.49 ± 0.35	5.42 ± 0.38	5.91 ± 0.93	5.79 ± 0.84	6.15 ± 1.24	6.22 ± 1.30	<0.001	<0.001
Diabetes mellitus, n (%)	0(0.0)	0(0.0)	220(24.5)	120(18.2)	233(30.1)	186(29.6)	<0.001	<0.001
Hypoglycaemic drugs, n (%)	0(0.0)	0(0.0)	165(18.4)	91(13.8)	177(22.8)	143(22.8)	<0.001	<0.001
Weight, kg	82.81 ± 13.32	70.09 ± 12.78	82.67 ± 13.06	69.59 ± 13.82	83.40 ± 13.20	70.87 ± 12.90	0.505	0.228
Height, cm	171.74 ± 6.98	158.47 ± 6.78	170.11 ± 7.02	157.19 ± 6.65	169.64 ± 7.11	156.64 ± 6.61	<0.001	0.006
BMI, (kg/m^2^)	28.05 ± 3.84	27.98 ± 5.20	28.54 ± 3.93	28.22 ± 5.65	28.92 ± 3.76	28.88 ± 4.91	0.003	0.028
WC, cm	99.98 ± 10.52	91.79 ± 12.37	101.77 ± 10.71	93.18 ± 13.22	102.70 ± 9.62	96.26 ± 12.47	<0.001	<0.001
Obesity, n (%)	71(24.1)	52(28.9)	272(30.3)	212(32.1)	248(32.0)	231(36.8)	0.040	0.072
Arterial stiffness								
ba-PWV, m/s	11.91 ± 1.17	11.08 ± 1.45	13.59 ± 1.43	13.25 ± 1.99	16.52 ± 2.49	16.18 ± 2.67	<0.001	<0.001

Values are means ± standard deviations for continuous data and number and proportions for categorical data. HVA: healthy vascular aging; NVA: normal vascular aging; EVA: early vascular aging; MD: Mediterranean diet; SBP: systolic blood pressure; DBP: diastolic blood pressure; LDL: low-density lipoprotein; HDL: high-density lipoprotein; FPG: fasting plasma glucose; HbA1c: glycosylated hemoglobin; BMI: body mass index; WC: waist circumference; *p*-value: differences between groups.

**Table 4 nutrients-16-04229-t004:** Association between ba-PWV and the MD overall and by sex determined by multinomial logistic regression.

Global		OR	(CI 95%)	*p*
MD				
	HVA vs. NVA	1.751	(1.411 to 2.174)	<0.001
	HVA vs. EVA	1.501	(1.295 to 1.740)	<0.001
Women				
MD				
	HVA vs. NVA	2.055	(1.456 to 2.901)	<0.001
	HVA vs. EVA	1.413	(1.124 to 1.776)	0.003
Men				
MD				
	HVA vs. NVA	1.551	(1.175 to 2.047)	0.002
	HVA vs. EVA	1.549	(1.275 to 1.882)	<0.001

Multinomial logistic regression using ba-PWV as dependent variables. The MD was used as an independent variable. Age, sex and consumption of antihypertensive drugs, hypoglycemic drugs and lipid-lowering agents were used as adjustment variables. MD: Mediterranean diet; HVA: healthy vascular aging; NVA: normal vascular aging; EVA: early vascular aging; OR: odds ratio; CI: confidence interval.

## Data Availability

The data supporting the findings of this study are available on ZENODO under the doi https://doi.org/10.5281/zenodo.12166167.

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
