# Peer review of "Effect of Dietary Patterns on Vascular Aging Using the Brachial–Ankle Index"

_nutrients, 2024, doi:10.3390/nu16234229_

Round 1
Reviewer 1 Report
Comments and Suggestions for Authors
Overall evaluation:
This research focuses on mediterranean diet (MD) and vascular aging, and the research program is basically reasonable. The research results have certain significance for the promotion of Mediterranean diet and the prevention and treatment of cardiovascular diseases. The research results are innovative, but the results are presented in a simple form, and the analysis and discussion of the results of this paper need to be strengthened.
Specific modification suggestions
1, Line2, delete "Title", the first letter of some words in the title need to be capitalized.
2, Line31, the introduction of the research background is somewhat inappropriate. A large number of studies have confirmed that the mediterranean diet (MD) can delay vascular aging, please modify it reasonably.
3, Line54, proposes a definition of "arterial stiffness".
4. Due to the lack of existing research progress in the introduction, it is suggested to supplement mediterranean diet (MD) and summarize relevant studies on vascular aging.
5, Line91, in addition to nutrition and exercise, psychological balance is also an important measure of anti-aging.
6, 2.4.2. Mediterranean diet, the total energy requirements of the daily diet should be stated.
7, Line231-234, the note should be adjusted to the lower part of the title.
8. In Figure 2, add the ordinate name to the figure.
9. The presentation of research results is relatively simple, so it is suggested to consider the correlation analysis of different results and make heat maps or cluster analysis charts to improve the level of the paper.
10. The discussion content is not deep enough and the content is small, so the reasons for this study should be discussed in depth and compared with similar studies.
11. Conclusions should indicate how to apply the results of this study to public health disease prevention.
Author Response
Comentarios de los revisores: Respuesta de los autores
Revisor #1:
Abrir reseña
() No me gustaría firmar mi informe de revisión
(X) Me gustaría firmar mi informe de revisión
Calidad del idioma inglés
(x) La calidad del inglés no limita mi comprensión de la investigación.
( ) El inglés podría mejorarse para expresar más claramente la investigación.
|
|
Sí |
Se puede mejorar |
Hay que mejorarlo |
No aplicable |
¿La introducción proporciona suficientes antecedentes e incluye todas las referencias relevantes? |
|
|
X |
|
¿Es adecuado el diseño de la investigación? |
|
X |
|
|
¿Están adecuadamente descritos los métodos? |
X |
|
|
|
¿Se presentan claramente los resultados? |
|
X |
|
|
¿Las conclusiones están respaldadas por los resultados? |
|
X |
|
|
Comentarios y sugerencias para los autores
Esta investigación se centra en la dieta mediterránea (DM) y el envejecimiento vascular, y el programa de investigación es básicamente razonable. Los resultados de la investigación tienen cierta importancia para la promoción de la dieta mediterránea y la prevención y tratamiento de las enfermedades cardiovasculares. Los resultados de la investigación son innovadores, pero los resultados se presentan de forma sencilla, y es necesario fortalecer el análisis y la discusión de los resultados de este trabajo.
En primer lugar, queremos agradecerle la revisión del manuscrito, así como los comentarios sugeridos, que mejorarán su contenido y comprensibilidad.
Sugerencias de modificación específicas
1, Línea2, elimine "Título", la primera letra de algunas palabras en el título debe escribirse en mayúscula.
1.- Hemos eliminado la palabra Título y la hemos modificado en el sentido que nos indiques. En la versión actual se ve así:
"Efecto de los patrones dietéticos en el envejecimiento vascular utilizando el índice braquial-tobillo".
2, Line31, la introducción de los antecedentes de investigación es algo inapropiada. Un gran número de estudios han confirmado que la dieta mediterránea (DM) puede retrasar el envejecimiento vascular, por favor modifíquela razonablemente.
2- Gracias por su comentario, hemos cambiado la introducción de los antecedentes de investigación según su indicación.
"La dieta mediterránea (DM) juega un papel importante en el retraso del envejecimiento vascular".
3, Line54, propone una definición de "rigidez arterial".
3.- Hemos incluido una definición de rigidez arterial en la línea 59 y la versión actual dice lo siguiente:
La rigidez arterial es la resistencia de las arterias a la deformación en respuesta a los cambios de presión/flujo con cada latido del corazón. La rigidez arterial es un marcador no invasivo de etapas tempranas a tardías de la aterosclerosis y se considera un sustituto de la enfermedad cardiovascular [1].
- Due to the lack of existing research progress in the introduction, it is suggested to supplement mediterranean diet (MD) and summarize relevant studies on vascular aging.
4.- Thank you very much for your comment. The information on the mediterranean diet and vascular aging has been completed.
5, Line91, in addition to nutrition and exercise, psychological balance is also an important measure of anti-aging.
5.- As the reviewer comments, both nutrition and exercise play an important role in aging and are supported by several studies [2-4]. Similarly, psychological balance probably plays an important role. However, in this case not all studies have shown this relationship [5]. On the other hand, we have not collected in this study the psychological state of all subjects. For this reason, we cannot analyze this effect.
We have reflected these aspects in the introduction in the new version as follows:
Good nutrition [4], lifestyles [5], physical activity [2,3] and psychological balance [6] are the pillars that help to have a healthy aging [4].
6, 2.4.2. Mediterranean diet, the total energy requirements of the daily diet should be stated.
Thank you very much for your comment, unfortunately the Mediterranean Diet Adherence Screener (MEDAS) questionnaire only evaluates the type of food and the minimum or maximum number of times you eat per day or week to meet or not the different criteria, it does not evaluate the calories ingested, therefore this variable has not been taken into account and it is not possible to obtain it from this questionnaire.
7, Line231-234, the note should be adjusted to the lower part of the title.
We have made the change indicated by the reviewer.
- In Figure 2, add the ordinate name to the figure.
We have added the name in the ordinate axis.
- The presentation of research results is relatively simple, so it is suggested to consider the correlation analysis of different results and make heat maps or cluster analysis charts to improve the level of the paper.
Thank you very much for your comment, we have added a graphical representation of the correlations with a heat map to improve the quality of the article.
- The discussion content is not deep enough and the content is small, so the reasons for this study should be discussed in depth and compared with similar studies.
Thank you very much for your comment, we have completed the discussion as you requested. We hope you like it.
- Conclusions should indicate how to apply the results of this study to public health disease prevention.
The results of this study indicate that greater adherence to MD is associated with healthier vascular aging. This suggests the importance of promoting MD as a public health intervention strategy to prevent vascular aging and its consequences. Therefore, health policy strategies are needed to promote the Mediterranean style diet to the entire population. All this will help to reduce the burden on health care systems, preventing the development of chronic diseases and improving vascular aging.
- Elosua-Bayes, M.; Marti-Lluch, R.; Garcia-Gil, M.D.M.; Camos, L.; Comas-Cufi, M.; Blanch, J.; Ponjoan, A.; Alves-Cabratosa, L.; Elosua, R.; Grau, M., et al. Association of Classic Cardiovascular Risk Factors and Lifestyles With the Cardio-ankle Vascular Index in a General Mediterranean Population. Rev Esp Cardiol (Engl Ed) 2018, 71, 458-465, doi:10.1016/j.rec.2017.09.011.
- Bangsbo, J.; Blackwell, J.; Boraxbekk, C.J.; Caserotti, P.; Dela, F.; Evans, A.B.; Jespersen, A.P.; Gliemann, L.; Kramer, A.F.; Lundbye-Jensen, J., et al. Copenhagen Consensus statement 2019: physical activity and ageing. Br J Sports Med 2019, 53, 856-858, doi:10.1136/bjsports-2018-100451.
- Eckstrom, E.; Neukam, S.; Kalin, L.; Wright, J. Physical Activity and Healthy Aging. Clin Geriatr Med 2020, 36, 671-683, doi:10.1016/j.cger.2020.06.009.
- Mazza, E.; Ferro, Y.; Pujia, R.; Mare, R.; Maurotti, S.; Montalcini, T.; Pujia, A. Mediterranean Diet In Healthy Aging. J Nutr Health Aging 2021, 25, 1076-1083, doi:10.1007/s12603-021-1675-6.
- Gómez-Sánchez, M.; Gómez-Sánchez, L.; Patiño-Alonso, M.C.; Cunha, P.G.; Recio-Rodríguez, J.I.; Alonso-Domínguez, R.; Sánchez-Aguadero, N.; Rodríguez-Sánchez, E.; Maderuelo-Fernández, J.A.; García-Ortiz, L., et al. Envejecimiento vascular y su relación con estilos de vida y otros factores de riesgo en la población general española: Estudio de Envejecimiento Vascular Precoz. J Hypertens 2020, 38, 1110-1122, doi:10.1097/hjh.00000000000002373.
- Lorenzo, E.C.; Kuchel, G.A.; Kuo, C.L.; Moffitt, T.E.; Diniz, B.S. La depresión mayor y las características biológicas del envejecimiento. Ageing Res Rev 2023, 83, 101805, doi:10.1016/j.arr.2022.101805.

Reviewer 2 Report
Comments and Suggestions for Authors
I would like to suggest several modifications to enhance its clarity and rigor:
TITLE / ABSTRACT
- The study design (cross-sectional study) is not clearly stated and should be explicitly mentioned.
INTRODUCTION
- Lines 51–52: Include relevant epidemiological data to provide context.
METHODS
- The study should adhere to a reporting guideline, such as those available at EQUATOR Network. Authors should specify the guideline used in the study design and provide the completed checklist separately to the Editor.
- The section “2.3 Study Population” should be revised to include details about the population and setting, with a more comprehensive description of the study setting. “2.3 Study Population and Setting”
RESULTS
- The authors have presented the results clearly, with well-structured tables that effectively support the findings.
DISCUSSION
- Expand this section by addressing the themes that emerged in greater depth. Discuss the findings in relation to other studies, particularly those focusing on dietary patterns in the Spanish population (e.g., DOI: 10.3390/nu16091278).
- Additionally, consider discussing how comorbid conditions may influence the outcomes examined in this study.
CONCLUSIONS
- Broaden this section by including potential future implications and practical applications for clinical practice.
ADDITIONAL CONSIDERATIONS
The article is well-structured and organized, with distinct sections that comprehensively address the topic. The references cited are relevant and authoritative.
Overall, this manuscript provides a significant, albeit descriptive, contribution to the literature on this topic. My compliments to the authors for their work.
Author Response
Revisor 2.-
Open Review
(X )I would not like to sign my review report
() I would like to sign my review report
Quality of English Language
(x) The quality of English does not limit my understanding of the research.
( ) The English could be improved to more clearly express the research.
|
|
Yes |
Can be improved |
Must be improved |
Not applicable |
Does the introduction provide sufficient background and include all relevant references? |
|
X |
|
|
Is the research design appropriate? |
X |
|
|
|
Are the methods adequately described? |
|
X |
|
|
Are the results clearly presented? |
X |
|
|
|
Are the conclusions supported by the results? |
|
|
X |
|
Comments and Suggestions for Authors
Thank you very much for your comments and for taking the time to review our manuscript. We are sure that it has helped us a lot to improve the quality of the article.
I would like to suggest several modifications to enhance its clarity and rigor:
TITLE / ABSTRACT
- The study design (cross-sectional study) is not clearly stated and should be explicitly mentioned.
We have added the following sentence in the abstract:
The main objective of this study was to analyze the association between adherence to the MD and vascular aging estimated with ankle brachial pulse wave velocity (ba-PWV) in a Spanish population sample and the differences by sex. Methods: Cross-sectional descriptive study. A total of 3437 subjects from the EVA, MARK and EVIDENT studies participated.
INTRODUCTION
- Lines 51–52: Include relevant epidemiological data to provide context.
Thank you for your comment. We have added this information as per your request. If you want us to add any more changes, please, tell us and we will do it.
“Cardiovascular diseases continue to be the leading cause of death worldwide, with an estimated 24 million deaths per year by 2030”.
Blais C, Rochette L, Ouellet S, Huynh T. Complex Evolution of Epidemiology of Vascular Diseases, Including Increased Disease Burden: From 2000 to 2015. Can J Cardiol. 2020 May;36(5):740-746. doi: 10.1016/j.cjca.2019.10.021. Epub 2019 Oct 25. PMID: 32146067.
METHODS
- The study should adhere to a reporting guideline, such as those available at EQUATOR Network. Authors should specify the guideline used in the study design and provide the completed checklist separately to the Editor.
- The section “2.3 Study Population” should be revised to include details about the population and setting, with a more comprehensive description of the study setting. “2.3 Study Population and Setting”
We have completed section 2.3 in the new version as follows:
The three studies were conducted in primary care centers. The three studies included 4103 subjects between 35 and 75 years of age. In this work 669 subjects have been excluded because not all the variables necessary to carry out this work have been recorded. Data from 3437 subjects were analyzed. In the MARK study [23], by random sampling of subjects consulting in 6 primary care centers in three autonomous communities, a total of 2511 individuals recruited were between 35 and 75 years of age and had an intermediate cardiovascular risk as a 10-year coronary risk between 5% and 15% estimated with the adapted Framingham risk equation [21], a 10-year vascular mortality risk between 1% and 5% estimated with the equation for the European population [22], or a moderate risk according to the European Society of Hypertension criteria for the treatment of arterial hypertension [23]. Exclusion criteria included the following: having end-stage disease, being institutionalized at the time of the visit, or having a history of atherosclerosis. Recruitment was performed from July 2011 to June 2013. Including in the analysis of this work 2505 subjects.
- Marrugat, J.; D’Agostino, R.; Sullivan, L.; Elosua, R.; Wilson, P.; Ordovas, J.; Solanas, P.; Cordón, F.; Ramos, R.; Sala, J.; et al. An adaptation of the Framingham coronary heart disease risk function to European Mediterranean areas. J. Epidemiol. Commun. Health 2003, 57, 634–638. [CrossRef]
- Conroy, R.M.; Pyörälä, K.; Fitzgerald, A.P.; Sans, S.; Menotti, A.; De Backer, G.; De Bacquer, D.; Ducimetière, P.; Jousilahti, P.; Keil, U.; et al. Estimation of ten-year risk of fatal cardiovascular disease in Europe: The SCORE project. Eur. Heart J. 2003, 24, 987–1003. [CrossRef]
- Mancia, G.; Fagard, R.; Narkiewicz, K.; Redón, J.; Zanchetti, A.; Böhm, M.; Christiaens, T.; Cifkova, R.; De Backer, G.; Dominiczak, A.; et al. 2013 ESH/ESC Guidelines for the management of arterial hypertension: The Task Force for the management of arterial hypertension of the European Society of Hypertension (ESH) and of the European Society of Cardiology (ESC). J. Hypertens. 2013, 31, 1281–1357. [CrossRef] [PubMed]
The EVIDENT study [24] had 1104 subjects selected by random sampling from a primary care center. Exclusion criteria were known coronary or cerebrovascular atherosclerotic disease; heart failure; moderate or severe chronic obstructive pulmonary disease; musculoskeletal disease that limited walking; advanced respiratory, renal, or hepatic disease; severe mental disease; treated oncological disease diagnosed in the 5 years. Recruitment was performed from January 2014 to September 2016. Including in the analysis of this work 432 subjects.
In the EVA study [25], using random sampling with replacement stratified by age groups (35, 45, 55, 65 and 75 years) and sex, we selected 501 subjects without previous cardiovascular disease receiving care in 5 urban health centers. We selected 100 subjects in each group, half of each sex, between 35 and 75 years of age from a reference population of 43 946 included in the individual health card database. Inclusion criteria were age between 35 and 75 and absence of previous cardiovascular disease. Exclusion criteria: terminally ill subjects, inability to travel to health centers, history of cardiovascular disease, glomerular filtration rate < 30 ml/min/1.73 m2, chronic inflammatory disease or an acute inflammatory process in the last 3 months, or being on estrogen, testosterone, or growth hormone treatment. Recruitment was performed from June 2016 to November 2017. Including in the analysis of this work 500 subjects.
Figure 1 shows the subjects included and excluded from each of the 3 studies.
RESULTS
- The authors have presented the results clearly, with well-structured tables that effectively support the findings.
We thank you for your comments and for reviewing the manuscript.
DISCUSSION
- Expand this section by addressing the themes that emerged in greater depth. Discuss the findings in relation to other studies, particularly those focusing on dietary patterns in the Spanish population (e.g., DOI: 10.3390/nu16091278).
- Additionally, consider discussing how comorbid conditions may influence the outcomes examined in this study.
Thank you for your comment. We have completed the information in the discussion section as per your request.
CONCLUSIONS
- Broaden this section by including potential future implications and practical applications for clinical practice.
Thank you for your comment. We have modified the conclusion section as per your request.
The results of this study indicate that greater adherence to MD is associated with healthier vascular aging. This suggests the importance of promoting MD as a public health intervention strategy to prevent vascular aging and its consequences. Therefore, health policy strategies are needed to promote the Mediterranean style diet to the entire population. All this will help to reduce the burden on health care systems, preventing the development of chronic diseases and improving vascular aging.
ADDITIONAL CONSIDERATIONS
El artículo está bien estructurado y organizado, con distintas secciones que abordan el tema de manera integral. Las referencias citadas son relevantes y autorizadas.
En general, este manuscrito proporciona una contribución significativa, aunque descriptiva, a la literatura sobre este tema. Mis felicitaciones a los autores por su trabajo.
Les agradecemos todos y cada uno de los comentarios realizados en la reseña, que sin duda mejorarán la comprensión de la misma.

Round 2
Reviewer 2 Report
Comments and Suggestions for Authors
The authors have made appropriate changes to the manuscript. It can now be published